# Liver Growth Factor “LGF” as a Therapeutic Agent for Alzheimer’s Disease

**DOI:** 10.3390/ijms21239201

**Published:** 2020-12-02

**Authors:** Rafael Gonzalo-Gobernado, Juan Perucho, Manuela Vallejo-Muñoz, Maria José Casarejos, Diana Reimers, Adriano Jiménez-Escrig, Ana Gómez, Gonzalo M. Ulzurrun de Asanza, Eulalia Bazán

**Affiliations:** 1Servicio de Neurobiología, Instituto Ramón y Cajal de Investigación Sanitaria (IRYCIS), 28034 Madrid, Spain; rd.gonzalo@cnb.csic.es (R.G.-G.); juan.perucho@outlook.com (J.P.); manuela.vmqm@gmail.com (M.V.-M.); m.jose.casarejos@hrc.es (M.J.C.); diana.reimers@hrc.es (D.R.); adriano.jimenez@hrc.es (A.J.E.); go.s6506@hotmail.com (A.G.); gonzalo.munoz@hrc.es (G.M.U.d.A.); 2National Centre for Biotechnology (CNB), CSIC, 28049 Madrid, Spain; 3Servicio de Neurología, Hospital Ramón y Cajal, 28034 Madrid, Spain

**Keywords:** liver growth factor, Alzheimer’s disease, inflammation, neuroprotection, microglia, Tg2576 transgenic mice, amyloid-beta

## Abstract

Alzheimer’s disease (AD) is a progressive degenerative disorder and the most common cause of dementia in aging populations. Although the pathological hallmarks of AD are well defined, currently no effective therapy exists. Liver growth factor (LGF) is a hepatic albumin–bilirubin complex with activity as a tissue regenerating factor in several neurodegenerative disorders such as Parkinson’s disease and Friedreich’s ataxia. Our aim here was to analyze the potential therapeutic effect of LGF on the APPswe mouse model of AD. Twenty-month-old mice received intraperitoneal (i.p.) injections of 1.6 µg LGF or saline, twice a week during three weeks. Mice were sacrificed one week later, and the hippocampus and dorsal cortex were prepared for immunohistochemical and biochemical studies. LGF treatment reduced amyloid-β (Aβ) content, phospho-Tau/Tau ratio and the number of Aβ plaques with diameter larger than 25 µm. LGF administration also modulated protein ubiquitination and HSP70 protein levels, reduced glial reactivity and inflammation, and the expression of the pro-apoptotic protein Bax. Because the administration of this factor also restored cognitive damage in APPswe mice, we propose LGF as a novel therapeutic tool that may be useful for the treatment of AD.

## 1. Introduction

Alzheimer’s disease (AD), the most common cause of dementia, was discovered by Alois Alzheimer in 1901. The prevalence of AD is about 1% in individuals aged 60–64, but shows an exponential increase with age, so that in people aged 85 years or older the prevalence is between 24–33% in the Western World. The hallmarks of the disease are extracellular deposits of plaques composed of amyloid-β (Aβ) and intracellular neurofibrillary tangles, composed of hyperphosphorylated tau (phospho-tau), a normal axonal protein that binds to microtubules [1]. The relationship among Aβ deposits, tangle formation and neurodegeneration, and cell death is not clear at present. The areas more vulnerable for the neurodegeneration processes are the cortex and the hippocampus, affecting mechanisms of spatial, semantic and episodic memories [2]. Other processes implicated in the pathological pathways in AD are cholinergic dysfunction, neuroinflammation, calcium channels, oxidation, iron chelation, abnormalities in the mitochondrial DNA and lipid metabolism, among others [3].

Therapeutic strategies in AD trying to ameliorate or eliminate its main disturbances, include the elimination of Aβ deposits, vaccination and immunization against Aβ, inhibitors of β or γ-secretases (enzymes excising Aβ from APP), antifibrillisation agents, statins (inhibitors of cholesterol synthesis), neuroprotectors, antioxidants and anti-inflammatory drugs, among others (reviewed in [4]). However, although the pathological hallmarks of AD are well defined, no effective therapy exists currently.

Liver growth factor (LGF) is a hepatic mitogen purified in 1986 by Dr. Diaz-Gil’s group [5]. This 64 kDa factor is an albumin–bilirubin complex that stimulates the proliferation of different cell types, and promotes regeneration of damaged tissues including the brain (reviewed in [6]). Thus, administration of LGF was able to stimulate axonal growth in the striatum, to increase the number of dopaminergic neurons in the damaged substantia nigra, and to improve rotational behavior stimulated by apomorphine in experimental Parkinson’s disease [7,8]. LGF also promoted the proliferation and migration of neural progenitors from the forebrain subventricular zone [9], and improved the viability, differentiation and integration of stem cell grafts into the host tissue [10]. Besides, our previous studies have reported the remarkable anti-inflammatory and antioxidant activities of LGF in extracerebral tissues [11,12,13], and in several experimental models of neurodegeneration where its main cellular target in the brain appears to be microglia [7,9,14,15]. Both effects are considered to be closely related with Alzheimer’s disease pathology [16,17].

To analyze the potential therapeutic effectiveness of LGF in AD, we have used Tg2576 transgenic mice (from now APPswe mice) that over-express the Swedish APP mutation (K670N/M671L) on a C57Bl/6 9 SJL background [18]. From 12 months onwards, these mice show Aβ plaque depositions in the hippocampus and cerebral cortex, reactive gliosis, inflammation and cognitive impairment that are neuropathological features of AD [19]. Chronic LGF administration to a 20–21-month-old APPswe mouse, significantly reduces Aβ and phospho-Tau protein levels and modulates protein ubiquitination and the expression of the heat shock protein 70 (Hsp70) that is involved in Aβ clearance. LGF also reduces microglia activation, astrogliosis and the expression of the apoptosis-associated speck-like protein containing a CARD (ASC), which is a component of the inflammatory response and cognitive impairment in AD pathology. Moreover, LGF up-regulates the expression of the Nuclear factor erythroid 2-related factor 2 (Nrf2), which plays an important role against oxidative stress. Because these beneficial effects correlate with a better cognitive outcome in APPswe mice, we may propose LGF as a potential novel therapeutic tool that may be useful for the treatment of AD.

## 2. Results

### 2.1. LGF Improves Behavioral Working Memory in APPswe Mice

The alternance index in the Y-maze, a parameter considered as an indicator of mnemonic function, was significantly reduced in the APPswe mice with respect to WT (Figure 1A). This important marker of working memory was recovered in APP-LGF treated mice (Figure 1A). APP mice also exhibited a reduced number of entries in comparison with WT mice, indicating a low general locomotor activity (18.8 ± 0.9 (*n* = 10) and 11 ± 1.8* (*n* = 11) in WT and APP mice, respectively * *p* ≤ 0.05 vs. WT). Instead, no significant differences were observed in the number of entries after LGF treatment in comparison with APP mice (APP-LGF: 13 ± 1.7 (*n* = 12)).

The marble-burying test is a useful model of neophobia, anxiety, and obsessive-compulsive behavior. As shown in Figure 1B, the marble burying test index was significantly reduced in APPswe aged mice, and LGF recovered this index of potential hippocampal affection.

### 2.2. LGF Modulatesβ-Amyloid Protein Expression in Hippocampus and Cerebral Cortex of APPswe Mice

β-amyloid protein accumulation in the Central Nervous System (CNS) is the most important feature of the experimental model of AD used in this study. As shown in Figure 2A, APPswe mice expressed the APP 695 human protein, and several Aβ peptides of different molecular weight lower than 60 kDa which expression was up-regulated in the hippocampus and cerebral cortex, in comparison with WT mice (Figure 2B). The immunohistochemical analysis also revealed the presence of Aβ-positive plaques in the hippocampus (40 ± 7 (*n* = 5) Aβ-positive plaques/mm^2^) and cerebral cortex (38 ± 6 (*n* = 5) Aβ-positive plaques/mm^2^) of APPswe mice. These plaques had different sizes with a 78 ± 9% of them showing a diameter lower than 25 µm in both brain structures (Figure 2C). In the APP-LGF experimental group, the total number of plaques was not significantly different to that observed in APPswe mice (54.6 ± 3.2 (*n* = 4) and 44 ± 3.5 (*n* = 4) Aβ-positive plaques/mm^2^in the hippocampus and cerebral cortex, respectively). However, LGF treatment significantly reduced Aβ protein levels (Figure 2B), and the number of plaques with a diameter higher than 25 µm compared to APPswe mice treated with the vehicle (Figure 2D). Besides, LGF slightly augmented, but not significantly, augmented the percentage of plaques with a diameter lower than 25 µm by 1.5 ± 0.1-fold and 1.3 ± 0.14-fold in the hippocampus and cerebral cortex, respectively.

### 2.3. Effects of LGF in Tau Phosphorylation and Protein Ubiquitination

A neuropathological feature of AD is the accumulation of Tau and ubiquitin in the neurofibrillary tangles (NFT). At 20 months of age, phospho-Tau/Tau ratio was increased by 2.5-fold in the hippocampus of APPswe mice, and LGF administration partially reduced this parameter to similar values of those observed in WT mice (Figure 3A,D). Phospho-Tau/Tau ratio was similar in the cerebral cortex of APPswe and WT mice, but LGF treatment significantly reduced this ratio below WT levels (Figure 3A). A similar effect was observed in the cerebral cortex of WT-LGF treated mice (Figure 3A).

Protein ubiquitination is essential in an important number of processes including protein degradation by the proteosome. Accumulation of polyubiquitinated proteins was observed in the hippocampus and cerebral cortex of APPswe mice, and LGF administration reduced this parameter in both structures (Figure 3B,D).

HSP-70 is a chaperone that allows the correct folding of mis-folded proteins that could lead to their aggregation. As shown in Figure 3C, the hippocampus of APP-LGF treated mice showed significantly higher levels of HSP-70 than WT or APPswe mice. HSP-70 protein expression was down-regulated in the cerebral cortex of APPswe mice, and LGF treatment restored its levels to control values (Figure 3C,D).

### 2.4. LGF Modulates Microglia Activation and Reduces Inflammation in APPswe Mice

Recent reports suggest that microglia play an important role in the evolution of AD. The immunohistochemical analysis of the hippocampus and cerebral cortex of APPswe mice showed that the number of microglial cells that expressed the Ionized Calcium Binding Protein-1 (Iba1) was significantly higher than in WT control mice (Figure 4A). Most of the Iba1-positive cells associated with the plaques were located around them (Figure 4B), and a few showed Iba1-positive processes penetrating into the plaques (Figure 4C). Iba1 protein levels were also up-regulated in both structures (Figure 4D,H), suggesting the presence of activated microglia in these mice. Increased proliferation has also been associated with microglia activation. As shown in Figure 4E, the expression of PCNA was significantly increased in the hippocampus and cerebral cortex of APPswe mice. Similar to this finding, BrdU incorporation was higher in these mice than in the WT group in both structures (Figure 4F), and more than 61 ± 4% (*n* = 8) of the proliferating cells were Iba1-positive microglia (Figure 4G).

The APP-LGF group of animals showed a significant reduction in Iba1 (Figure 4D,H) and PCNA protein expression (Figure 4E,H), and a slight but reduced number of BrdU-positive cells in comparison with APPswe mice (Figure 4F). Besides, LGF treatment increased from 6 ± 2.9 (*n* = 3) to 19 ± 11 (*n* = 4) the percentage of Iba1-positive cells, which prolongations penetrated profoundly in the plaques in the hippocampus.

GFAP is a protein expressed by astrocytes for which up-regulation has been associated with inflammatory states in chronic processes as AD and aging. As shown in Figure 5A, GFAP was over-expressed in the hippocampus and cerebral cortex of APPswe mice, and LGF treatment significantly reduced its levels in both structures. On the other hand, LGF decreased ASC protein levels that were up-regulated in the hippocampus and the cerebral cortex of APPswe aged mice (Figure 5B,D). ASC is involved in the production of Interleukin-1beta (Il-1beta) but, neither APP nor APP-LGF treated mice showed any significant change in the levels of this pro-inflammatory cytokine in the hippocampus (113 ± 6 (*n* = 6) and 101 ± 7 (*n* = 6) % of IL-1beta expression vs. WT in APP and APP-LGF treated mice, respectively) and the cerebral cortex (102 ± 11 (*n* = 5) and 92 ± 12 (*n* = 6) % of IL-1beta expression vs. WT in APP and APP-LGF treated mice, respectively). Similar results were observed when we analyzed the potential effects of LGF in Tumor Necrosis Factor-alpha (TNF-alpha) protein expression in the hippocampus (92 ± 6 (*n* = 6) and 88 ± 10 (*n* = 8) % of TNF-alpha expression vs. WT in APP and APP-LGF treated mice, respectively) and cerebral cortex (95 ± 6 (*n* = 5) and 107 ± 9 (*n* = 5) % of TNF-alpha expression vs. WT in APP and APP-LGF treated mice, respectively). This latter cytokine mediates LGF-induced neuroregeneration/neuroprotection [6,20].

LGF also modulated the expression of the transcription factor Nrf2, and the class B scavenger receptor CD36, two proteins that regulate oxidative stress and inflammatory responses in AD. As shown in Figure 5D, Nrf2 protein levels were reduced in the hippocampus and cerebral cortex of APPswe mice, and LGF treatment up-regulated Nrf2 protein expression to reach control values. LGF treatment reduced the levels of CD36 that is over-expressed in the hippocampus of APPswe mice (Figure 5C,E). Besides, LGF up-regulated CD36 protein expression in the hippocampus of WT mice (Figure 5C,E), but it lacked activity in the cerebral cortex, a brain structure where neither APP nor APP-LGF mice showed any alteration in CD36 protein levels (Figure 5C).

### 2.5. LGF Modulates the Expression of Proteins Involved in Cell Survival

Our previous studies showed how in an experimental model of Parkinson’s disease, LGF potentiates cell survival through the regulation of Bcl2 and Bax protein expression [9]. In the hippocampus of APPswe mice the Bcl2/Bax ratio was significantly reduced, and LGF treatment increased this ratio to control values (Figure 6A). This effect was due to a reduction in the expression of the pro-apoptotic protein Bax which levels were significantly up-regulated in the hippocampus of APPswe mice (Figure 6B).

Akt is another protein involved in cell survival modulated by LGF in neurodegeneration [9]. As shown in Figure 6C, phospho-Akt/Akt ratio was up-regulated in the hippocampus of APPswe mice where phospho-Akt levels were significantly increased in comparison with WT mice (Figure 6D). The APP-LGF group of mice also showed an increased phospho-Akt/Akt ratio and higher levels of phospho-Akt than control mice (Figure 6C,D). Neither Bcl2/Bax nor phospho-Akt/Akt ratio were affected in the cerebral cortex at any of the experimental conditions used in this study (Figure 6A,B).

## 3. Discussion

In the experimental model of AD used in this study (APPswe mouse), LGF treatment reduced Aβ and phospho-Tau protein levels as well as the number of Aβ plaques with a diameter higher than 25 µm. LGF administration also modulated protein ubiquitination and HSP70 protein levels, reduced glial reactivity and inflammation, and up-regulated the expression of proteins involved in cell survival. These LGF beneficial effects correlate with a better cognitive outcome in APPswe mice.

Amyloid-β aggregation and deposition of Aβ plaques are considered as causative agents in AD, so many therapies have been directed to modify Aβ levels (reviewed in [21]). Under our experimental conditions, LGF modulated Aβ by reducing its levels to WT values in the hippocampus and cerebral cortex. This reduction was observed in the expression of those Aβ peptides with a molecular weight lower than 60 kDa. This fraction includes the Aβ1-42 peptide that has been extensively studied due to its abundance in patients predisposed to AD [22] and mouse models of AD [19]. Here, we also show that LGF treatment reduced the accumulation of polyubiquitinated proteins, and up-regulated HSP70 protein expression in both structures. The high levels of ubiquitinated proteins observed in APPswe mice indicate that proteins are not correctly proteolyzed, probably due to the generation of free radicals and/or an inflammatory environment promoted by the accumulation of Aβ HSP70 is a chaperone which overexpression could suppress the progression of AD by promoting Aβ clearance (reviewed in [23]). HSP70-mediated Aβ clearance from the brain has been associated with the up-regulation of insulin-degrading enzyme (IDE) and transforming growth factor β1 (TGF-β1) [24]. This study did not evaluate IDE and TGF-β1 expression but LGF administration significantly reduced plasmatic levels of insulin in an experimental model of atherosclerosis in mice [25], and modulated the synthesis of TGF-β1 in bile duct-ligated rats [11].

Another major histopathological characteristic in AD is neurofibrillary tangles composed of aggregates of hyperphosphorylated forms of the protein associated with Tau microtubules [26,27]. As shown here, LGF treatment reduced phospho-Tau/Tau ratio in the hippocampus of APPswe mice. Interestingly, Tau homeostasis is controlled through the action of molecular chaperones, such as HSP70 [28,29] that, as noted above, was up-regulated in this structure. In the case of the cerebral cortex, phosphoTau/Tau ratio in APP mice was not significantly different from untreated WT mice but, in this brain structure, LGF treatment reduced phospho-Tau/Tau ratio in both, WT and APP mice. At present we do not have a possible explanation for this event. We could suggest that this is a potential specific effect of LGF not associated to the accumulation of Aβ plaques and inflammation observed in APP mice. However, according to our results it does not seem the case because similar results were not observed in the hippocampus of WT mice where, as already mentioned, LGF did not reduced p-Tau/Tau ratio. Besides HSP70, which was also up-regulated in the cerebral cortex of APP-LGF mice, LGF could modulate Tau phosphorylation in this brain structure by different mechanisms not analyzed in this study. Further studies must be conducted in order to elucidate these discrepancies.

Activated microglia and reactive astrocytes are two pathological markers observed in AD [30,31,32]. Both cell types are considered cell targets of LGF in the CNS [6,7,9,20]. According to the present results, LGF significantly reduced proliferation (as measured by BrdU incorporation and PCNA protein expression) and Iba1 protein levels, which are two features of activated microglia. These results are in agreement with previous studies that show how chronic administration of LGF reduced microglia cell reactivity in an experimental model of Parkinson’s disease in rats [9] and in ataxic rats [15]. Our immunohistochemical studies also suggest that the administration of LGF could prime microglia to acquire a phagocytic phenotype. Thus, although the number of Iba1-positive cells was not reduced in APP-LGF treated mice, this treatment increased by 3-fold the percentage of Iba1-positive cells that invaded Aβ plaques and reduced the number of Aβ large plaques while increasing the number of smaller plaques, which would explain the slight, but not significant increase observed in this parameter. How LGF modulates microglia activity in APPswe mice is unknown but, as mentioned above, LGF could stimulate TGFβ1expression that is a cytokine that stimulates Aβ clearance through activation of phagocytic microglia [33]. On the other hand, activation of phagocytic microglia has been associated with increased levels of interleukin 6 and TNF-alpha [34]. This latter cytokine seems to play an important role in the mitogenic cascade of LGF in the liver, and in LGF-induced neuroregeneration (reviewed in [6]) and neuroprotection [20].

Reactive astrogliosis is a well-known hallmark of AD that is identified, among others, by an increased expression of GFAP. We also found that LGF reduced GFAP over-expression in the hippocampus and cerebral cortex of APPswe mice, suggesting that astrocytes are potential targets of LGF in this experimental model of AD. Microglia cell reactivity to Aβ and phagocytic activity are also under the regulation of astrocytes [35] so, their stimulation with LGF could lead to the release of factors with antioxidant and/or anti-inflammatory effects that could modulate both events. In this regard, we should mention that LGF modulates the expression of TNF-alpha in cultures of mesencephalic glia [20], and significantly reduced by 36 ± 11% the expression of ASC in these cultures (*p* ≤ 0.05 vs. untreated cultures).

The anti-oxidative and anti-inflammatory activities of LGF have also been observed in vivo. Thus, chronic LGF decreased oxidized glutathione concentration in ataxic mice [14], and reduced TNF-alpha protein levels in 6-OHDA-lesioned rats [9]. As presented here, LGF modulates the expression of several proteins that regulate oxidative stress and inflammatory responses in AD. ASC is an adaptor protein that participates in the production of pro-inflammatory cytokines, and contributes to the inflammatory response and cognitive impairment in AD pathology [36,37,38,39]. Chronic administration of LGF down-regulated ASC protein levels in the hippocampus and cerebral cortex of APPswe mice. In these mice, ASC is mainly expressed by microglia, which is associated with Aβ plaques (Reimers et al. manuscript in preparation). Because ASC released by microglia forms an aggregate complex with Aβ that promotes inflammation [37,39], we may argue that ASC reduction is associated with a potential anti-inflammatory effect of LGF in this experimental model of AD. Although we were unable to detect any significant change in TNF-alpha and Il-1beta protein expression at the experimental time analyzed in this study, we cannot exclude the possibility that LGF modulates the expression of these cytokines at earlier times of treatment, as observed for TNF-alpha in 6-OHDA-lesioned rats [9]. In addition, small changes in the levels of both proteins may not be detected by Western blot analysis, so that it may be necessary to use other methods with higher sensitivity as ELISA. Other studies have reported that reduced levels of ASC stimulate phagocytosis in glial cells in a mouse model of AD [40], so LGF-induced down-regulation of ASC could also mediate the increase in phagocytic activity proposed in our study.

CD36 is a class B scavenger receptor involved in phagocytosis which expression in microglia plays a key role in the Aβ-mediated activation of this cell type (reviewed in [41]). Moreover, CD36 over-expression in microglia seems to mediate the inflammatory processes associated to Aβ pathology in transgenic mice [42], and the production of ROS in response to Aβ in AD human brains [43,44]. Similarly, the increased levels of CD36 observed in the hippocampus of APPswe mice could contribute to creating an inflammatory environment in this structure. According to our results, LGF treatment could modulate inflammation and/or ROS production in the APPswe hippocampus by decreasing CD36 protein expression and, in consequence, its interaction with Aβ in this structure. In fact, the inhibition of Aβ interaction with CD36 has been proposed as a potential therapeutic strategy in AD [45]. An interesting observation of our study is that LGF up-regulated CD36 levels in the hippocampus of WT mice. These high levels of CD36 could promote the phagocytic activity of microglia to improve the clearance of debris that is accumulated in the normal brain during the processes of aging [46,47]. Although similar results were observed in the cerebral cortex of APPswe and APP-LGF treated mice, no statistically significant differences were detected. Perhaps CD36 levels are not sufficient to be detected through Western blot in this structure, or we need to study a higher number of mice, or mice at different ages. De facto, we were unable to detect CD36 up-regulation in the hippocampus of thirteen months old APPswe mice (100 ± 7 (*n* = 4) and 104 ± 6 (*n* = 4) % of CD36 expression in WT and APPswe mice, respectively).

Nrf2 is a transcription factor that regulates oxidative stress [48,49] and anti-inflammatory responses [50,51]. Nrf2 mRNA and protein deficiency have been observed in AD patients [52,53] and AD models in mice [54,55]. As shown here, LGF up-regulated Nrf2 protein levels that were reduced in the cerebral cortex and the hippocampus of APPswe mice. Nrf2 up-regulation ameliorated cognitive impairment in APP Knock-in AD mice [56], and in APPswe/PSEN1dE9 hypoxic mice, the intranasal administration of Nrf2 modulated CD36 protein expression, reduced Aβ deposition, and improved spatial memory [57]. In both studies, these beneficial effects were associated to a reduction in oxidative stress and neuroinflammation promoted by the increased levels of Nrf2. Since in our mice LGF significantly ameliorates cognitive impairment, we may consider that up-regulation of Nrf2 could underlie this behavioral improvement through the activation of anti-oxidant and anti-inflammatory responses, as has been reported in other experimental pathologies[58].

Our previous results indicate a potential neuroprotective role of LGF in the brainstem and cerebellum of ataxic rats [15], and in the substantia nigra and striatum of 6-hydroxydopamine-lesioned rats [9]. In both experimental models of neurodegeneration, LGF raised the ratio Bcl2/Bax due to the up-regulation of the anti-apoptotic protein Bcl2, which is involved in the survival of neural cells[59]. In the experimental model of AD used in this study, LGF also restored Bcl2/Bax ratio to control values, but this effect was due to the down-regulation of the pro-apoptotic protein Bax that was over-expressed in the hippocampus of APPswe mice. Up-regulation of Bax has been observed in neurons that exhibit neurofibrillary tangle pathology [60], and APP transgenic mice [61]. Besides, a neuroblastoma cell line co-transfected with Swedish mutant APP showed a higher Bax/Bcl2 ratio [62]. LGF-induced down-regulation of Bax could contribute to decreasing the excessive neuronal apoptosis reported in AD [63]. The increase in Bax levels observed in APPswe could result in neuronal death. However, APPswe mice did not show any reduction in the expression of the specific nuclear neuronal marker NeuN in the hippocampus (100 ± 8 (*n* = 6) and 99 ± 10 (*n* = 5) % of NeuN expression in WT and APP mice, respectively), and cerebral cortex (100 ± 7 (*n* = 6) and 99 ± 12 (*n* = 5) % of NeuN expression in WT and APP mice, respectively). These results agree with the lack of neuronal death previously reported in this experimental model of AD [64,65]. Akt, is a serine/threonine kinase which phosphorylated form plays a critical role in the regulation of neuronal survival [66,67] that has been involved in the pathogenesis of AD [68]. Our results showed an increased phospho-Akt/Akt ratio in the APPswe hippocampus due to the higher levels of phospho-Akt observed in this brain structure. Phospho-Akt may protect neuronal cell death through different mechanisms including the phosphorylation and inhibition of cytosolic pro-apoptotic proteins (reviewed in [69]). Because the phosphorylation of Bax by active Akt affects its translocation to the mitochondria [70], we may consider the up-regulation of phospho-Akt in the hippocampus as an intrinsic mechanism to protect neurons of Bax-mediated apoptotic cell death in the APPswe mice.

From our study, we cannot define if the observed effects of LGF are due to a direct or indirect action of the factor on the brain of APPswe mice. A potential direct action would imply that LGF could cross the blood–brain barrier by binding to specific receptors found in the endothelium known as RAGE receptors (receptors for advanced glycation end-products) [71]. In fact, LGF show some structural similarities with the advanced glycation end products (AGE) which structure is mainly based on albumins bound to glucose with the ability to change albumin conformation [72]. RAGE receptors are also expressed by astrocytes and microglia [73], and their activation elicits TNF-alpha release from these cells [73].

Present results show how LGF treatment also improves memory performance impairment in APPswe mice. The APPswe mice in comparison with the WT animals, had a reduced ambulatory behavior in the Y-maze, as shown by the total number of entrances and this pattern was unchanged by LGF treatment, so we may consider that motor activity and anxiety were affected in this experimental model of AD. However, the alternance index in the Y-maze, which is an indicator of mnemonic function, was significantly reduced in these mice. LGF treatment recovered this important parameter of working memory to similar values to those observed in WT mice. Other studies have reported a reduction in the exploratory behavior from 10 months old APPswe mice that are coincident with the presence of Aβ plaque depositions in the hippocampus and cerebral cortex (reviewed in [74]). Thus, LGF-induced reduction in Aβ plaques size and levels could contribute to the cognitive improvement observed in our study. In addition, LGF could ameliorate the exploratory behavior in these mice through the reduction of glial reactivity, and its anti-inflammatory and anti-oxidative responses referred to above.

The marble-burying test index was lower in the APPswe mice than in WT mice, and LGF rescued this behavioral parameter. Other studies have shown reduced scrabble behavior and buried marbles in mice with cortical and hippocampal lesions [75] and in Tg-APP/PS1 mice [76]. In the latter study, the behavioral deficits were attributed to the accumulation of Aβ in the brain, so we may consider that the beneficial effects observed in APP-LGF treated mice could be in part due to the reduction in Aβ levels promoted by the factor.

## 4. Materials and Methods

### 4.1. Ethics Statement

The Ethics Committee of the Hospital Ramón y Cajal, Madrid (animal facilities ES280790002001) approved on 11 May 2015 all the protocols related to the use of laboratory animals. All procedures associated with animal experiments were in accordance with Spanish legislation (RD 53/2013) and the European Union Council Directive (2010/63/EU).

### 4.2. LGF Purification

LGF was obtained and purified from serumof rats that were subjected to a bile duct ligation for 5 weeks, as previously described by Díaz-Gil et al. [77]. The quantitation of the purity of LGF was performed by HPLC [78] and serum samples that showed the highest concentrations of the growth factor were chosen to continue with the process. The method included three chromatographic procedures using DEAE-cellulose, Sephadex G-150, and hydroxylapatite. The absence of contaminants and other growth factors (purity) in the LGF samples was also evaluated according to the standard criteria described by Díaz-Gil et al. [5,77,79]. All LGF preparations showed a single band in SDS-PAGE electrophoresis and were lyophilized and stored at 4 °C until use. Finally, LGF aliquots were dissolved in saline prior to intraperitoneal administration.

### 4.3. Transgenic Animals

The Tg2576 line over-expressing human APP695 was used as a model of AD in this study. This line contains the human mutated APPswe that includes the double mutation M671L and K670N under the control of the prion protein promoter [18]. This mouse line of AD was generously donated by Dr. Carro [80]. These mice develop age-dependent AD-type neuropathology [18]. Tg2576 mice show elevated levels of the major soluble form of brain amyloid (Aβ1-40) at 1 year old. An intercross between APP Tg2576 and Wild-type (WT) C57BL6 was performed in order to obtain WT littermate controls.

Transgenic (Tg) animals were generated by breeding the mice according to the following diagram: male APPswe × female WT → APPswe (50/57.6) and WT (50/42.4)—(the numbers in brackets show the expected/found genotype frequencies of the offspring expressed in %).

Eighteen WT mice were divided into 2 experimental groups, WT control (WT, *n* = 11) and LGF treated group (WT-LGF, *n* = 7). APPswe mice were also divided in APPswe control (APP, *n* = 12) and LGF treated APPswe (APP-LGF, *n* = 16). Four mice per group were used for histological assays, and 4 to 10 mice were used for biochemical studies. All mice included in each experimental group were used for behavioral studies. The animals were studied at 21 months of age.

### 4.4. Genotype by PCR

The genotype of each animal was confirmed as described by Carro et al. [80]. Genomic DNA was extracted from the mouse tail using the High Pure PCR template preparation kit (Roche, Barcelona, Spain) according to the manufacturer’s instructions. Briefly, 150 ng of DNA was amplified by polymerase chain reaction (PCR) using the specific primers described in [80] to detect mutant human APP sequence.

### 4.5. LGF and BrdU Administration

Twenty-month-old mice received 2 weekly 100 µL i.p. injections of saline used as a vehicle (WT and APP groups) or LGF (1.6 μg/100 µL) (WT-LGF and APP-LGF groups) for 3 weeks. This optimal dose of LGF has been used in different model systems using an identical or similar schedule [13,14]. Mice were sacrificed 1 week after the last treatment with vehicle or LGF (Figure 7).

To determine whether the i.p. administration of LGF increased cell proliferation, a group of WT and APP animals were injected daily for 3 weeks with the mitotic marker 5-bromodeoxyuridine (BrdU) (50 mg/kg i.p.), starting 24 h after the beginning of LGF/saline injections.

### 4.6. Behavioral Studies

Exploratory behavior and body weight were measured before treatment initiation to allocating the animals to the experimental groups (WT, WT-LGF, APP, APP-LGF). After 3 weeks of treatment with vehicle or LGF, mice were subjected to behavioral analysis. Two behavioral tests were carried out in this study: Y-maze test and marble burying test.

#### 4.6.1. Y-Maze

Spatial working memory was measured using the Y-maze behavioral paradigm. The Y-maze test is based on the natural exploratory preference of mice to alternate arms when exploring a new environment. The spontaneous alternation behavior of mice in the Y-maze was analyzed to assess short term memory impairment in mice, following the methodology previously reported by Perucho et al. [81,82]. Testing takes place in a Y-shaped maze with three white, opaque plastic arms spaced at an angle of 120 degrees. Briefly, after acclimatizing the mice to the behavioral room (30 min), each mouse was placed at the same end of one arm of the Y-maze and allowed to move freely through the maze for 5 min. The number of arms entered, as well as the number of spontaneous alternations, defined as a sequence of entries in each of 3 consecutive arms without repetition, were recorded to determinate the alternation rate. The percentage of alternation score was calculated as follows:(1)$$\mathrm{Spontaneous alternation}\%=\left(\frac{\mathrm{number of spontaneous alternations}}{\mathrm{total number of arm entries}-2}\right)\times 100$$

#### 4.6.2. Marble Burying Test

Marble burying was assessed as an additional measure of exploratory digging. This test is a useful paradigm to study repetitive compulsive-like behaviors based on the observation of rodents burying objects in their bedding, a phenomenon dependent on hippocampal function [75]. Briefly, mice were individually housed in a cage filled with 5 cm of wood chip bedding for a 30 min testing period of acclimation, and continued throughout the test session. Tests were performed between 18:00 and 20:00 h, under standard laboratory conditions of temperature, 22 ± 2 °C, 12 h light:dark cycle, relative humidity 50–60% and with food and water restriction. The mice were placed individually for 30 min in plastic cages, with 9 marbles placed on top of the bedding material, in a 3 × 3 grid with 4 cm center-to-center spacing. A new cage, clean marbles and fresh bedding were used for each mouse. At the end of that time, each mouse was returned to its home cage and the number of marbles uncovered; completely buried (not visible); 2/3 covered (only top visible) and half covered, were counted. This classification was used to obtain a “buried index”, as an indication of activity and interaction with their environment [83].

### 4.7. Antibodies and Immunochemicals for Immunohistochemistry

The primary antibodies used in this study were: rabbit polyclonal anti-Iba1 (1:100; Wako Chemicals USA, Inc., Los Angeles, CA, USA), mouse monoclonal anti-5-bromodeoxyuridine (BrdU, 1:25; DakoCytomation, Glostrup, Denmark), and mouse monoclonal anti-Aβ (6E10, 1:100; Covance, Emeryville, CA, USA). The secondary antibodies used were: Alexa Fluor-568 goat anti-mouse IgG, and Alexa Fluor-488 goat anti-rabbit IgG (1:400; both from Molecular Probes; Eugene, OR, USA), fluorescein-conjugated goat anti-mouse IgG (1:25; Jackson ImmunoResearch Laboratories Inc., West Grove, PA, USA).

### 4.8. Tissue Processing, Immunohistochemistry and Morphometric Analyses

Four weeks after the beginning of LGF or saline treatment transgenic mice were perfused intracardially under deep anesthesia with 10 mL of heparinized isotonic saline, followed by 40 mL of 4% paraformaldehyde. Brains were postfixed in the same solution for 24 h at 4 °C, cryoprotected and frozen, before sectioning into 20 µm-thick coronal sections in a cryostat. For quantitative measurements and immunohistochemical analysis of the dorsal hippocampus and dorsal cortex, coronal sections were obtained at the antero-posterior level of −2.5 to −3 mm from Bregma [84].

Tissue sections were mounted on coated FLEX IHC microscope slides (DakoCytomation, Glostrup, Denmark), treated with sodium acetate 10 mM, pH 6.0, at 95 °C for 4 min, and blocked with 5% normal goat serum (NGS) and 0.1% Triton-X 100 in phosphate buffer saline (PBS), pH 7.4, for 30 min. Primary antibodies diluted in 0.5% NGS in PBS, were applied for 24 h at 4 °C, and were visualized using fluorescent secondary anti-rabbit and anti-mouse antibodies diluted in 0.5% NGS in PBS. The slides were coverslipped in a medium containing p-phenylenediamine and bisbenzimide (Hoechst 33342; Sigma, St. Louis, MO, USA) for detection of nuclei. For double immunolabeling with anti-Iba1 and anti-BrdU antibodies, the former immunostaining was performed prior to the immunodetection of BrdU incorporated into nuclei. The latter detection needed pretreatment of sections with 2N HCl at 37°C for 30 min, previous to the NGS blocking step. In the case of Aβ immunodetection in plaques, sections were pretreated with 70% formic acid for 20 min at RT, followed by the blocking step.

For quantification of proliferating BrdU-positive cells, microglial Iba1-positive cells, and Aβ 6E10-positive plaques, panoramic views were obtained from one representative coronal section for each immunostaining technique and for each brain. These panoramic views, where the complete transversal surface of dorsal hippocampus and dorsal cortex of each cerebral hemisphere were visualized (3 mm^2^ for each structure), were obtained by using the stitching tool of the software NIS ELEMENTS C, version BR 3.2 (Nikon Instruments Inc., Tokio, Japan), coupled to a Nikon ECLIPSE Ti-e microscope (Nikon Instruments Inc., Tokio, Japan) with a motorized stage, and the 10× objective. Four WT, 4 APP and 4 APPL-GF were analyzed for these evaluations. The data entered in the corresponding graphs were: (a) the mean ± SEM obtained from the data of both cerebral hemispheres (number of plaques/mm^2^); (b) the mean of 2 different regions in the cortex and of 4 different regions in the hippocampus of one cerebral hemisphere (density of Iba1-positive cells); and (c) the sum of data from both hemispheres (density of BrdU-positive cells).

### 4.9. Brain Regions and Tissue Preparation for Biochemical Analysis

After decapitation the brain was extracted and the hippocampus and cortex were free-hand dissected, then the samples were frozen on dry ice for biochemical studies (Western blot). For the protein extraction, dried tissue samples were weighed and placed in six volumes (w/v) of phosphate-buffered saline (PBS) with protease inhibitor cocktail 1× (Calbiochem, Temecula, CA, USA) and 20 mM N-ethylmaleimide to inactivate deubiquitinating enzymes and subjected to two 30-s rounds of sonication. The lysates were immediately boiled for 5 min and centrifuged at 12,000× *g* at 4 °C for 30 min. The supernatant-PBS-inhibitors was defined as the soluble fraction and was used for protein analysis by Western blot. To obtain the total Aβ (soluble plus insoluble) fraction, the initial homogenate (200 μL) was extracted with 5 M guanidine in 6 mM Tris–HCl, pH 8.0 by rotating the sample at room temperature overnight. The sample obtained after guanidine extraction represented the total fraction. Soluble and total fractions were used for protein analysis by Western blot.

### 4.10. Protein Analysis

Aliquots of 20–30 µg of protein were added to sample loading buffer 2× (50 mM Tris pH 6.8, 10% glycerol, 2% SDS, 5% β-mercaptoethanol and 0.1% bromophenol blue). The electrophoresis was performed using SDS-polyacrylamide gels (10–15%) and then the samples were electro-blotted using nitrocellulose membranes (0.45 µm). The blots were blocked using TTBS solution (5% dry skimmed milk, 137 mM NaCl plus 0.1% Tween-20 and 20 mM Tris-HCl pH 7.6) at room temperature for 2 h. Once the blocking of non-specific binding was performed, the membranes were incubated overnight at 4 **°**C with specific primary antibodies diluted in blocking solution. Then, the membranes were washed twice for 10 min using blocking solution followed by another two washes with TTBS for 5 min. β-actin was used as a loading control and housekeeping protein. The membranes were developed by enhanced chemiluminescence detection using a commercial kit (Bio-Rad Laboratories Inc., Hercules, CA, USA) and quantified by computer-assisted video densitometry using the Bio-Rad *Quantity One* software (Bio-Rad Laboratories Inc, Hercules, CA, USA). The data of the proteins of interest analyzed in the study were normalized with respect to β-actin levels.

#### 4.10.1. Primary Antibodies

The primary antibodies used in this study were: Mouse monoclonal amyloid-β antibody 6E10 (BioLegend Inc., San Diego, CA, USA), diluted 1:1000. Mouse monoclonal anti-HSP70 (1:1000) and rabbit polyclonal antibody against ASC (1:500) were from Santa Cruz (Temecula, CA, USA). Mouse monoclonal anti tau-5 (for measurement of total tau protein) antibody (Chemicon, Madrid, Spain) diluted 1:5000 and rabbit polyclonal to phospho-tau (phospho serine199 + serine202) (1:1000) were from Abcam (Cambridge, UK). Mouse monoclonal anti-glial fibrillar acid protein (GFAP) antibody diluted 1:5000 was from Chemicon (Madrid, Spain) and rabbit polyclonal antibody for anti-Iba1, 1:1000 was for WAKO (Japan). Rabbit anti-proliferating cell nuclear antigen (PCNA, 1:75; SantaCruz Biotechnology, Santa Cruz, CA, USA), and mouse monoclonal anti-ubiquitin antibody (1/500) was from Chemicon (Chemicon International Inc., Temecula, CA, USA). Mouse anti Bcl2 (1:250) and rabbit anti-Bax (1:250) were from Santa Cruz (Santa Cruz Biotechnology). Rabbit anti-phospho-Akt (Ser473P) (1:2000) and rabbit anti-Akt (1:2000) were from Cell Signaling Technology, Beverly, MA, USA. Rabbit anti Nrf2 (1:1000) and rabbit anti CD36 (1:500) were from Thermo Fisher Scientific, Rosemont, IL, USA. To correct and quantify the protein charge mouse monoclonal anti-β-actin antibody diluted 1:10,000 was used from Sigma.

#### 4.10.2. Secondary Antibodies

Goat anti-mouse-IGg-HRP and anti-rabbit-IGg-HRP secondary antibodies diluted 1:1000 were purchased from Bio-Rad (Bio-Rad Laboratories Inc., Hercules, CA, USA).

### 4.11. Statistical Analysis

Results are expressed as mean ± SEM of (*n*) independent animals. Statistical analysis was performed with the GraphPad Prism software, version 6.01 (La Jolla, CA, USA). Before analysis, the Shapiro–Wilk test was used to test normality. One-way ANOVA followed by the Newman–Keuls multiple comparison test were performed. Differences were considered significant when *p* ≤ 0.05.

## 5. Conclusions

Liver growth factor is an albumin–bilirubin complex that exhibits remarkable neuroprotective, anti-inflammatory, and anti-oxidant activities in several models of neurodegenerative diseases. In the experimental model of AD used in this study (APPswe mouse), chronic LGF treatment reduced Aβ and phospho-Tau protein levels which are the main neuropathological features associated to the disease. LGF also reduced microglia and astroglia cell reactivity and modulated the expression of ASC CD36 and Nrf2 which are proteins involved in the regulation of inflammatory and oxidative processes. Because these beneficial effects correlated with an improvement in the cognitive deficits observed in the APPswe mice, we may consider LGF as a potential new therapeutic factor for AD.

## 6. Patents

US 8,642,551 B2, 4 February 2014CE, num. 09732019.6-1456, Ref. EP-883, April 2015US 14/140.014, 26 May 2015

## Figures and Tables

**Figure 1 ijms-21-09201-f001:**
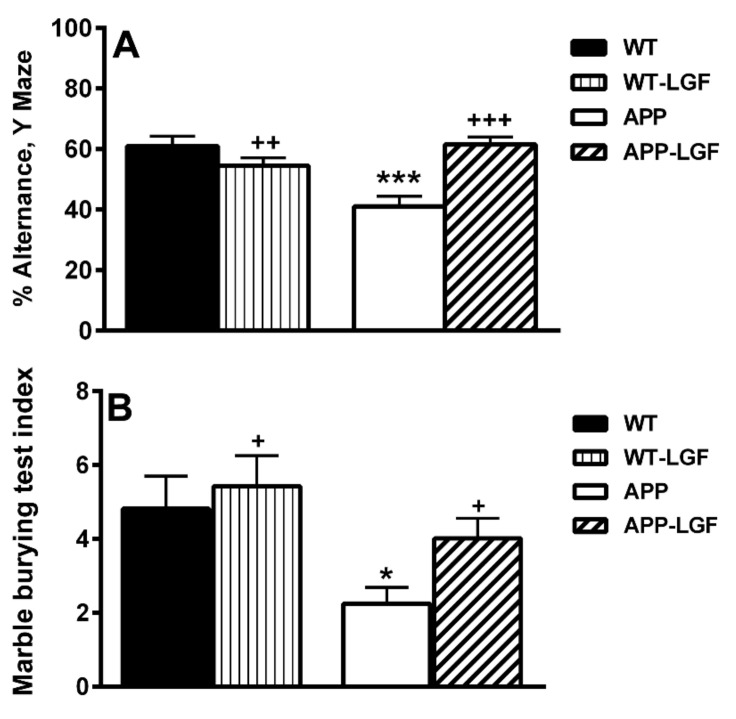
Liver growth factor (LGF) improves cognitive behavior in APPswe mice. Panel (**A**) shows how the alternance index in the Y-maze is significantly reduced in APP mice, and how LGF treatment recovers this important marker of working memory. As shown in panel (**B**), the marble burying test index is significantly reduced in APP mice, and LGF recovers this potential index of hippocampal and cortical affection. Results represent the mean ± SEM of 11 to 16 independent mice. The statistical analysis was performed by one-way ANOVA followed by Newman–Keuls test. * *p* ≤ 0.05, *** *p* ≤ 0.001 vs. wild type (WT). + *p* ≤ 0.01, ++ *p* ≤ 0.01, +++ *p* ≤ 0.001 vs. APP mice.

**Figure 2 ijms-21-09201-f002:**
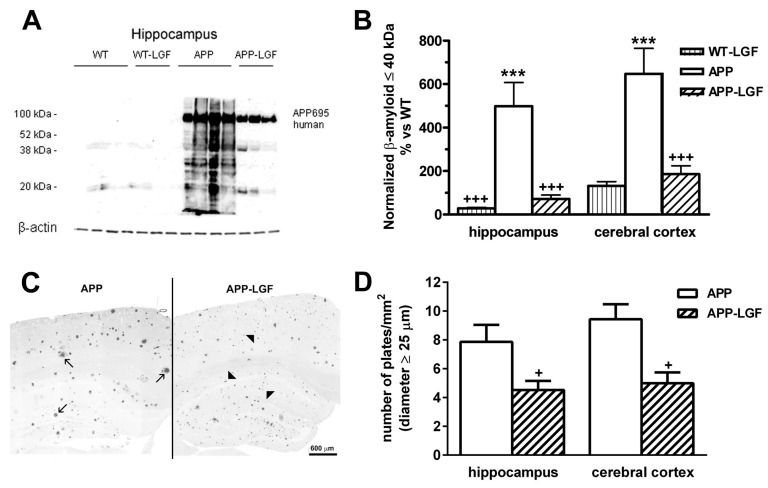
LGF reduces amyloid-β accumulation and the size of amyloid-β positive plaques in the hippocampus and cerebral cortex of APPswe mice. Panels (**A**,**B**) show a Western blot representative image of Aβ aggregates in the hippocampus (**A**) and the quantification of Aβ-aggregates ≤40 kDa in the hippocampus and cerebral cortex (**B**). Note that Aβ protein levels are up-regulatedin the hippocampus and cerebral cortex of APPswe mice and that LGF treatment significantly reduces its expression in both structures. Panel (**C**) shows a representative image of Aβ-positive plaques of large (**C**, black arrows) and small size (**C**, black arrowheads) in one hemisphere of APP mice brain (scale bar: 600 µm). Note that in APP-LGF mice, large plaques are reduced, while small plaques increase. Panel (**D**) shows how LGF treatment significantly reduces the number of plaques with a diameter higher than 25 µm in the hippocampus and cerebral cortex. Results represent the mean ± SEM of 7 to 8 (**B**) and 4 to 5 (**D**) independent mice in each experimental group. The statistical analysis was performed by one-way ANOVA followed by Newman–Keuls test. *** *p* ≤ 0.001 vs. WT. + *p* ≤ 0.05, +++ *p* ≤ 0.001 vs. APP mice.

**Figure 3 ijms-21-09201-f003:**
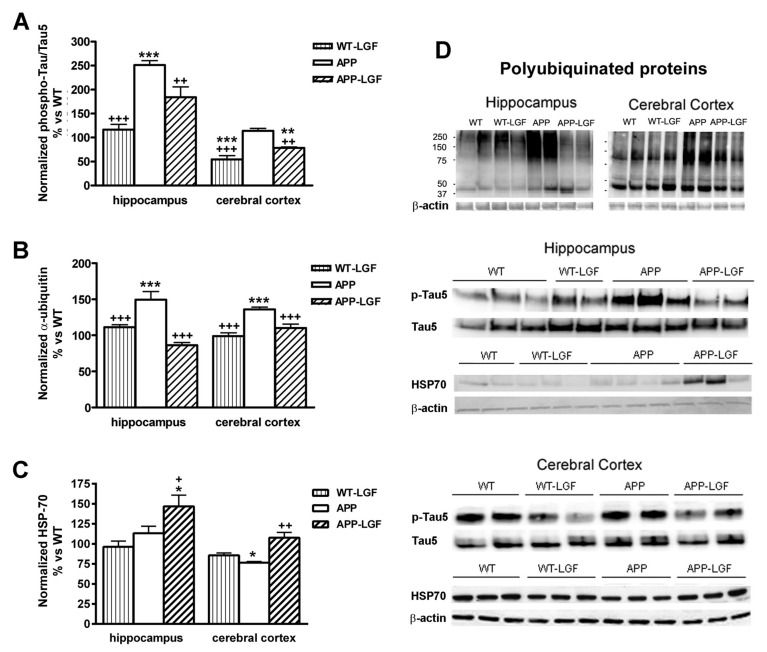
Effects of LGF in tau pathology, ubiquitinated protein levels and HSP70 heat shock protein expression. Panel (**A**), shows the relative optical density of phospho-tau/tau ratio analyzed by Western blot and expressed as % vs. WT. Note how LGF treatment reduces the expression of phosphorylated tau in the hippocampus and cerebral cortex in APP mice. Panels (**B**,**C**) show the quantification of accumulated ubiquitinated proteins (**B**) and HSP-70 (**C**) in the hippocampus and cerebral cortex of APPswe mice. Note how LGF reduces the accumulation of ubiquitinated proteins (**B**) and increases HSP70 expression (**C**). Panel (**D**) shows representative Western blot images of phospho-tau, total tau, Hsp-70 and β-actin as charge control. Results represent the mean ± SEM of 4 to 8 independent mice. The statistical analysis was performed by one-way ANOVA followed by Newman–Keuls test. * *p* ≤ 0.05, ** *p* ≤ 0.01, *** *p* ≤ 0.001 vs. WT. + *p* ≤ 0.05, ++ *p* ≤ 0.01, +++ *p* ≤ 0.001 vs. APPswe mice.

**Figure 4 ijms-21-09201-f004:**
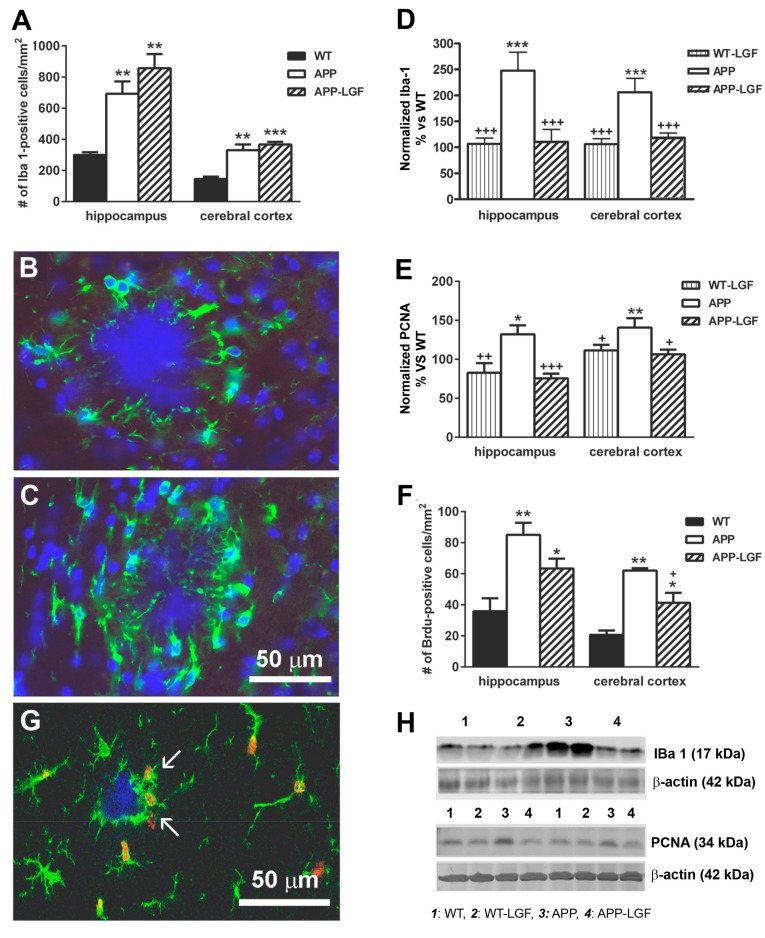
Liver growth factor modulates microglia cell activity in the hippocampus and cerebral cortex of APPswe mice. Panel (**A**) shows the quantification of Iba1-positive microglia cells in APPswe mice. Panels (**B**,**C**) show different responses of Iba1-positive cells: in (**B**) microglial prolongations do not penetrate inside the deposit, while in (**C**) they actively phagocytize the plaque (scale bar: 50 µm). Panels (**D**,**E**) show quantitative Western blot analysis of Iba1 (**D**) and PCNA (**E**) protein expression. Note how LGF administration significantly reduces the over-expression of Iba1 (**D**) and PCNA (**E**) in the hippocampus and cerebral cortex. LGF also reduces BrdU cell incorporation (**F**) as an index of cellular proliferation. Note how BrdU (**G**, red) is associated with Iba1-positive cells (**G**, green, white arrows). Panel (**H**) show representative Western blots of Iba1 and PCNA and their respective β-actin as charge control. The results represent the mean ± SEM of 4 (**A**,**F**) and 8 to 10 (**D**,**E**). independent mice. The statistical analysis was performed by one-way ANOVA followed by Newman–Keuls test. * *p* ≤ 0.05, ** *p* ≤ 0.01, *** *p* ≤ 0.001 vs. WT. + *p* ≤ 0.05, ++ *p* ≤ 0.01, +++ *p* ≤ 0.001 vs. APPswe mice.

**Figure 5 ijms-21-09201-f005:**
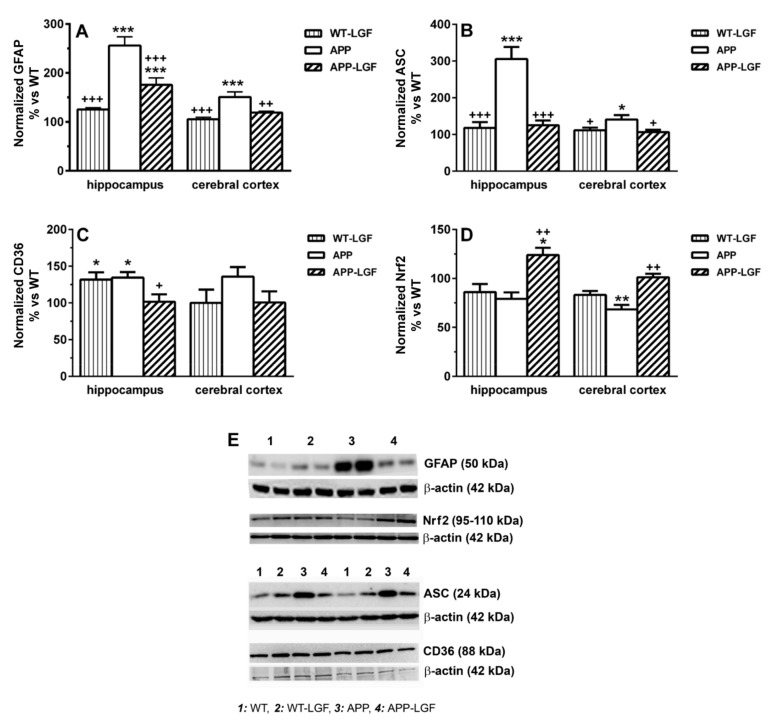
LGF reduces the inflammatory response in APPswe mice. Panel (**A**) shows how LGF significantly reduces GFAP protein expression in the cerebral cortex and hippocampus of APPswe. LGF administration also decreases the expression of ASC (**B**), which is a protein involved in the production of pro-inflammatory cytokines. Panel (**C**) shows quantitative Western blot analysis of the receptor scavenger CD36 that is over-expressed in the hippocampus of APP mice. Panel (**D**) represents the quantitative analysis of the transcription factor Nrf2 as an oxidative stress regulator. Representative Western blots of GFAP, Nrf2, ASC, CD36 and their respective β-actin as charge are shown in panel (**E**). Results represent the mean ± SEM of 4 to 10 independent mice. The statistical analysis was performed by one-way ANOVA followed by Newman–Keuls test. * *p* ≤ 0.05, ** *p* ≤ 0.01, *** *p* ≤ 0.001 vs. WT. + *p* ≤ 0.05, ++ *p* ≤ 0.01, +++ *p* ≤ 0.001 vs. APPswe mice.

**Figure 6 ijms-21-09201-f006:**
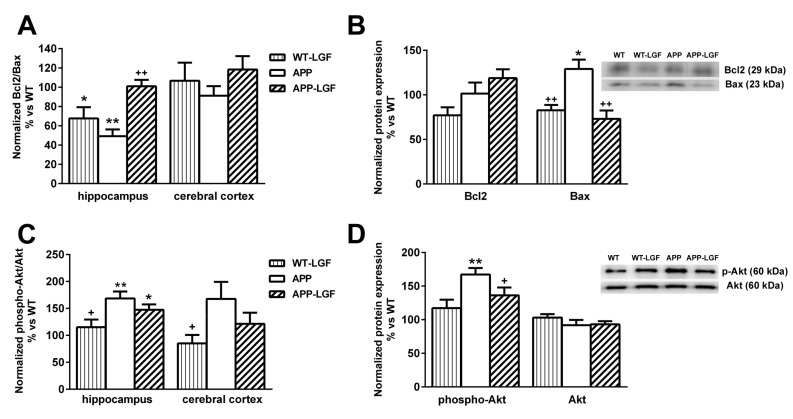
Effect of LGF on the expression of proteins involved in cell survival. Protein expression was analyzed by Western blot using specific antibodies against Bcl2, Bax, p-Akt and Akt. Panel (**A**), shows that LGF increases Bcl-2/Bax ratio in the hippocampus. This effect is mainly due to a reduction in Bax protein levels (**B**). Panels (**C**,**D**) show that phospho-Akt/Akt ratio (**C**) and phospho-Akt protein levels (**D**) are significantly increased in the hippocampus of APP and APP-LGF treated mice. Results represent the mean ± SEM of 4 to 8 independent mice. The statistical analysis was performed by one-way ANOVA followed by Newman–Keuls test. * *p* ≤ 0.05, ** *p* ≤ 0.01, vs. WT.+ *p* ≤ 0.05, ++ *p* ≤ 0.01 vs. APPswe mice.

**Figure 7 ijms-21-09201-f007:**
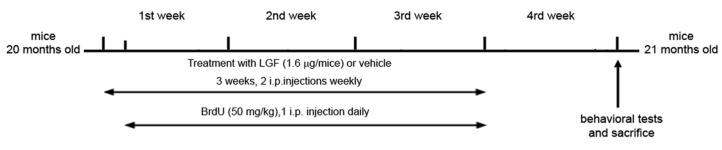
Representative scheme showing the methodological procedure followed for the administration of LGF or vehicle to 20 months old WT and APPswe mice.

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
