# Peer review of "Liver Growth Factor “LGF” as a Therapeutic Agent for Alzheimer’s Disease"

_ijms, 2020, doi:10.3390/ijms21239201_

Round 1
Reviewer 1 Report
The manuscript by Gonzalo-Gobernado and colleagues examine the effect of liver growth factor (LGF) on cognitive and molecular characteristics in a mouse model of AD. The authors demonstrate changes in a number of protein markers following i.p. injections of 1.6μg LGF twice a week for 3 weeks. Major concerns:
- It is unclear how the protein quantification was performed. Protein expression (for all proteins investigated) are reported as ‘protein x vs beta-actin’. It is unclear what ‘vs beta-actin’ refers to and the authors must explain how the quantification was performed. In the methods section of the manuscript, the authors state “β-actin was used as a loading control and housekeeping protein”. If the proteins were not normalized to the beta-actin signal then these results must be recalculated. The authors are referred to Analytical Biochemistry (2020) 593:113608 (doi.org/10.1016/j.ab.2020.113608).
- Section 4.4 is not required for this manuscript as it is a standard procedure for genotyping these mice. - While the entire manuscript requires language editing, section 4.5 and 4.6 were particularly poorly written. - Please avoid commencing a sentence with the word ‘And’ – see lines 189, 278 and 375.
- Ln 46 – immunization is against Abeta and not APP - Figure 2C – what is the arrowhead pointing to?
- Ln 124 – It is unclear what the authors mean by the following sentence – “Phospho-Tau/Tau ratio was not affected in the cerebral cortex of APPswe mice”
- Ln 132 – Please change “wrong-folded” to ‘mis-folded’.
- Figure 3A – This figure appears to show that LGF induces tau phosphorylation in WT mice. There appears to be an ~100% increase in the pTau:Tau5 ratio in the hippocampus and an ~ 50% increase in the cortex. The authors must explain this phenomenon in light of the fact that they state that LGF decreases pTau:Tau5 ratio in Tg mice.
- Figure 5E – The authors have numbered the lanes as follows Lane 1: WT, Lane 2: WT-LGF, Lane 3: APP, Lane 4: APP-LGF However, this only represents the bottom half of Figure 5E and not the top half (GFAP, Nrf2). In order to be inclusive, the authors could remove the word ‘Lane’ and leave the numbering as follows: 1: WT, 2: WT-LGF, 3: APP, 4: APP-LGF
- Line 339 - The significance of a re-emergence of a NeuN signal following LGF treatment is unclear.
- It is unfortunate that the authors did not use the Y-maze to test spatial memory in the treated mice but this is now beyond the possibility of this project. The authors should consider using more reliable and robust tests in future experiments.
Author Response
Observations to the comments of referee Number 1:
1. It is unclear how the protein quantification was performed. Protein expression (for all proteins investigated) are reported as ‘protein x vs beta-actin’. It is unclear what ‘vs beta-actin’ refers to and the authors must explain how the quantification was performed. In the methods section of the manuscript, the authors state “β-actin was used as a loading control and housekeeping protein”. If the proteins were not normalized to the beta-actin signal then these results must be recalculated. The authors are referred to Analytical Biochemistry (2020) 593:113608 (doi.org/10.1016/j.ab.2020.113608).
This information is now clearly included in the methods section (lines 594-597) as follows:
The membranes were developed by enhanced chemiluminescence detection using a commercial kit (Bio-Rad) and quantified by computer-assisted video densitometry using the Bio-Rad Quantity Onesoftware. The data of the proteins of interest analyzed in the study were normalized with respect to β-actin levels.
2. Section 4.4 is not required for this manuscript as it is a standard procedure for genotyping these mice. - While the entire manuscript requires language editing, section 4.5 and 4.6 were particularly poorly written. - Please avoid commencing a sentence with the word ‘And’ – see lines 189, 278 and 375
The word ‘And’ has been deleted from lines 189, 278 and 375.
Section 4.4 has been reduced in this version of the manuscript (lines 478 to 482). According to the reviewer suggestions, sections 4.5 (now lines 486 to 495) and 4.6 (now lines 499 to 533) have been modified in the new version of the manuscript. In addition, section 4.5 includes a figure (Fig. 7) to clarify the protocol followed in this study.
3. Ln 46 – immunization is against Abeta and not APP - Figure 2C – what is the arrowhead pointing to?
Line 46 has been changed by Ab
Fig. 2C legend has been also changed. In addition, and as suggested by the reviewer 2, now Figure 2C includes photomicrographs from APP and APP-LGF mice brains.
4. Ln 124 – It is unclear what the authors mean by the following sentence – “Phospho-Tau/Tau ratio was not affected in the cerebral cortex of APPswe mice”
In the cerebral cortex of APP mice Phospho-Tau/Tau ratio was not significantly different to that observed in WT mice. To avoid any confusion the sentence above referred has been changed as follows:
Phospho-Tau/Tau ratio was similar in the cerebral cortex of APPswe and WT
5. Ln 132 – Please change “wrong-folded” to ‘mis-folded’.
Ln 132 has been changed
6. Figure 3A – This figure appears to show that LGF induces tau phosphorylation in WT mice. There appears to be an ~100% increase in the pTau:Tau5 ratio in the hippocampus and an ~ 50% increase in the cortex. The authors must explain this phenomenon in light of the fact that they state that LGF decreases pTau:Tau5 ratio in Tg mice.
Phospho-tau/tau ratio was not modified by LGF in the hippocampus of WT mice in comparison with non treated WT mice. The significance observed in this experimental group (+++ symbol) was vs APP (vehicle) mice. Moreover, in the cerebral cortex phospho-Tau/Tau ratio was similar in WT and APP mice although LGF treatment significantly reduced this ratio in both, WT and APP mice. Trying to explain these differences we have included the following paragraph (lines 300 to 313):
In the case of the cerebral cortex, phosphoTau/Tau ratio in APP mice was not significantly different from untreated WT mice but, in this brain structure, LGF treatment reduced phospho-Tau/Tau ratio in both, WT and APP mice. At present we do not have a possible explanation for this event. We could suggest that this is a potential specific effect of LGF not associated to the accumulation of Aβ plaques and inflammation observed in APP mice. However, according to our results, it does not seem the case because similar results were not observed in the hippocampus of WT mice where, as already mentioned, LGF did not reduced p-Tau/Tau ratio. Besides HSP70 that was also up-regulated in the cerebral cortex of APP-LGF mice, LGF could modulate Tau phosphorylation in this brain structure by different mechanisms not analyzed in this study. Further studies must be conducted in order to elucidate these discrepancies.
7. Figure 5E – The authors have numbered the lanes as follows Lane 1: WT, Lane 2: WT-LGF, Lane 3: APP, Lane 4: APP-LGF However, this only represents the bottom half of Figure 5E and not the top half (GFAP, Nrf2). In order to be inclusive, the authors could remove the word ‘Lane’ and leave the numbering as follows: 1: WT, 2: WT-LGF, 3: APP, 4: APP-LGF
Lane has been deleted from Figures 4H and 5E
8. Line 339 - The significance of a re-emergence of a NeuN signal following LGF treatment is unclear.
Because we discuss about NeuN signal in lines 353 to 358 (now lines 399 to 404), we have excluded from line 339 (now line 390) any reference to the effects of LGF in NeuN expression observed in other studies. Thus, paragraph included in is written as follows:
In both experimental models of neurodegeneration, LGF raised the ratio Bcl2/Bax due to the up-regulation of the anti-apoptotic protein Bcl2that is involved in the survival of neural cells.
9. It is unfortunate that the authors did not use the Y-maze to test spatial memory in the treated mice but this is now beyond the possibility of this project. The authors should consider using more reliable and robust tests in future experiments.
One of the behavioral tests used in this study was Y-maze to test spatial working memory by measuring spontaneous alternation (Fig. 1A, in the results section). In our opinion, although there are other tests to asses spatial memory in mice (i.e. Morris water maze or Barnes Maze), the Y-maze is a simple, easy and reproducible test to perform. The Y-maze test can be also used in order to asses spatial reference memory, however this paradigm was not performed in this project. Unfortunately, we do not have adequate facilities in our institution to perform other behavioral test, but it is possible that in the near future we will be able to open a Central Unit specialized in the performance and analysis of different behavioral tests in rodents.
Reviewer 2 Report
The manuscript titled as ‘Liver Growth Factor “LGF” as a therapeutic agent for Alzheimer's disease’, by Gobernado et al., found within APP mice, liver growth factor (LGF) treatment educed amyloid-β (Aβ) content, phospho-Tau/Tau ratio and the number of A plaques with diameter larger than 25 µm. LGF administration also modulated protein ubiquitination and HSP70 protein levels, reduced glial reactivity and inflammation, and reduced the expression of the pro-apoptotic protein Bax. This work should be of wide interests to most researchers on pharmacology and neuroscience.
This manuscript has sufficient novel and findings and the method described is highly practical, however, the following points need to be addressed:
- The authors did not explain why use 'a dose of 1.6 μg 100 μl of LGF per mouse and day’ as a optimal dose for this study, was it based on their in vitro studies, or based on in vivo studies by others?
- Figure 2C, should include representative images from LGF treatment experiments.
- In Figure 3D, middle panel for Hippocampus, the blots for p-Tau5 and Tau5, were done in triplicates, however these were not matched with the blots for HSP70 and beta-Actin, which were done in duplicate? The same situation happens for those blots on Cerebral Cortex.
Author Response
Observations to the comments of referee Number 2:
1. The authors did not explain why use 'a dose of 1.6 μg 100 μl of LGF per mouse and day’ as a optimal dose for this study, was it based on their in vitro studies, or based on in vivo studies by others?
This dose and schedule have been previously tested in vivo. To avoid any confusion we have modified section 4.5 as follows:
Twenty-month-old mice received 2 weekly 100 µl IP injections of saline used as vehicle (WT and APP groups) or LGF [1.6 μg/100 µl] (WT-LGF and APP-LGF groups) for 3 weeks. This optimal dose of LGF has been used in different model systems using an identical or similar schedule [13,14]. Mice were sacrificed 1 week after the last treatment with vehicle or LGF.
2. Figure 2C, should include representative images from LGF treatment experiments.
As suggested by the reviewer, Figure 2C has been modified. Now it includes a panoramic view of one hemisphere of APP and APP-LGF mice brains. Figure 2C legend has been also modified.
We have also include in section 2.2 of the results data of the effects of LGF in total number of plaques and small plaques (lines 118 to 125)
3. In Figure 3D, middle panel for Hippocampus, the blots for p-Tau5 and Tau5, were done in triplicates, however these were not matched with the blots for HSP70 and beta-Actin, which were done in duplicate? The same situation happens for those blots on Cerebral Cortex.
We have not performed blots for p-Tau/Tau and HSP70 in the same gel. These proteins were performed in different gels in which we loaded the samples in a different way. Because P-Tau and Tau were developed in the same membrane, and in the results section we present their ratio, we do not consider necessary to show their respective beta-Actin bands. To avoid any confusion we have slightly modified Fig. 3D to include the different treatments in each respective blot
Round 2
Reviewer 1 Report
We thank the authors for addressing questions and comments made by the reviewers.
There are some minor edits that should be performed:
Ln 123 – Stating that “LGF slightly up-regulated the percentage of plaques with diameter lower than 25 μm by 1.5 ± 0.1-fold and 1.3 ± 0.14-fold in the hippocampus and cerebral cortex” suggests that LGF is adding to the plaque load. It is more likely that LGF is stimulating a degradative ‘factor’ that is actively degrading larger plaques and what is being observed are larger (>25mm) plaques that are in the process of being degraded (<25mm).
Figures – The ordinate axes in figures that report protein levels normalized to β-actin should be labelled as “Normalized ‘protein’ % vs WT” and not “’Protein’ vs β-actin % vs WT”
e.g., Normalized α-ubiquitin % vs WT
- please make these changes to all figures
Ln 447 – missing ‘of’ – “LGF was obtained and purified from serum OF rats….”
Ln 454 – replace ‘band’ with immunoreactive species’ and replace ‘4@C’ with ‘4oC’
Author Response
1.-Ln 123 – Stating that “LGF slightly up-regulated the percentage of plaques with diameter lower than 25 μm by 1.5 ± 0.1-fold and 1.3 ± 0.14-fold in the hippocampus and cerebral cortex” suggests that LGF is adding to the plaque load. It is more likely that LGF is stimulating a degradative ‘factor’ that is actively degrading larger plaques and what is being observed are larger (>25mm) plaques that are in the process of being degraded (<25mm).
As suggested by the reviewer it is possible that LGF could favor the activity of a single, or several, degrading factors that reduced the size of large Aβ plaques. How LGF modulates these factors is unknown, but our results suggest that LGF is able to prime microglia to acquire a phagocytic phenotype that could mediate the reduction of Aβ plaques through the synthesis and release of actively degrading factors. In this way LGF could reduce large plaques while increasing the number of smaller plaques, which would explain the slight, but not significant increase observed in this parameter. On the other hand, the amount of Aβ in this small plaques is probably reduced in APP-LGF mice vs their APP counterparts, as we deduce from the results of western blot analysis.
2.-Figures – The ordinate axes in figures that report protein levels normalized to β-actin should be labelled as “Normalized ‘protein’ % vs WT” and not “’Protein’ vs β-actin % vs WT”
e.g., Normalized α-ubiquitin % vs WT
- please make these changes to all figures
The ordinate axes of all figures have been changed
3.-Ln 447 – missing ‘of’ – “LGF was obtained and purified from serum OF rats….”
Of is been corrected in line 447
4.-Ln 454 – replace ‘band’ with immunoreactive species’ and replace ‘4@C’ with ‘4oC’
We appreciate very much that comment, but unfortunately we cannot replace in line 454 band by immunoreactive species. Due to the lack of antibodies against LGF, no immunostaining was performed during the purification of the factor, a silver-stain technique was performed and LGF preparations showed a single band of 64 kDa in SDS-PAGE electrophoresis. Authors only performed SDS-PAGE electrophoresis after the three chromatographic procedures commented in section 4.2. The protocol followed is detailed in the reference below indicated.
Díaz-Gil, J.J.; Escartin, P.; Garcia-Canero, R.; Trilla, C.; Veloso, J.J.; Sanchez, G.; Moreno-Caparros, A.; Enrique de Salamanca, C.; Lozano, R.; Gavilanes, J.G., et al. Purification of a liver DNA-synthesis promoter from plasma of partially hepatectomized rats. Biochem J 1986, 235, 49-55.
4@C is been corrected in line 454
Reviewer 2 Report
The authors have addressed most of my concerns and the manuscript has been improved, however, there is an issue on Figure 3, there is a part of the blotting data (the blots between APP and APP-LGF for HSP70) seems been manipulated (been cut, copied and pasted etc.).
The authors need to clarify this data, and repeat the experiment etc..

Author Response
1.-The authors have addressed most of my concerns and the manuscript has been improved, however, there is an issue on Figure 3, there is a part of the blotting data (the blots between APP and APP-LGF for HSP70) seems been manipulated (been cut, copied and pasted etc.).
The authors need to clarify this data, and repeat the experiment etc..
Blotting data presented for HSP70 in the hippocampus were obtained from a single gel and its corresponding membrane. Trying to reduce the number of lanes presented in the figure, we cropped some of the lanes. To avoid any confusion, in this new version of the manuscript we include the whole blotting membrane for HSP70 in hippocampus.
Round 3
Reviewer 2 Report
It is better now that authors provided the whole blotting membrance for HSP70 in Figure 3. The column graph for this blotting result is the same as previous version, I assume this was done with the whole blotting membreace, however, the standard deviation or ' ± SEM' for this part is not making sense, if you compare those ' ± SEM' of WT-LGF, or APP. The authors need to make sure the statistical analysis for this part is fine.
Author Response
1.- It is better now that authors provided the whole blotting membrane for HSP70 in Figure 3. The column graph for this blotting result is the same as previous version, I assume this was done with the whole blotting membrane, however, the standard deviation or ' ± SEM' for this part is not making sense, if you compare those ' ± SEM' of WT-LGF, or APP. The authors need to make sure the statistical analysis for this part is fine.
Results presented in Fig 3C represent the mean ± SEM of several independent mice that were performed in two gels. Fig. 3D shows one of these gels. Here, we include the other performed gel. If the reviewer considers that this second gel is more representative of the results presented in Fig. 3C for HSP70 in hippocampus, we do not have any problem in to modify it
We have also newly quantified both gels and their respective b-actine. We include here the mean ± SEM of the results obtained from this new quantification. Data for hippocampus if Figure 3C has been also modified according to the results obtained from this new quantification
WT= 100 ± 7 (N=5)
WT-LGF= 96,4 ± 7 (N=7)
APP= 113 ± 9 (N=7)
APP-LGF= 147 ± 14 (N=7) (*p ≤ 0.05 vs WT; + p ≤0.05 vs APP)
